# Harmonic oscillator based particle swarm optimization

**Yury Chernyak**[1]☣, **Ijaz Ahamed Mohammad**[1]☣*, **Nikolas Masnicak**[1],
**Matej Pivoluska**[1,2], **Martin Plesch**[1,3]☣

**1** Institute of Physics, Slovak Academy of Sciences, Bratislava, Slovakia, **2** QTlabs, Clemens-Holzmeister-Straße 6/6 Etage 6, Wien, Austria, **3** Matej Bel University, Národná ulica 12, Banská Bystrica, Slovakia

☣ These authors contributed equally to this work.

* fyziijaz@savba.sk

**Data availability statement:** All relevant data are within the manuscript.

**Funding:** MP acknowledge the support of VEGA project 2/0055/23 and projects 09I03-03-V04-00425 and 09I03-03-V04-00685 of the

## Abstract

Numerical optimization techniques are widely applied across various fields of science and technology, ranging from determining the minimal energy of systems in physics and chemistry to identifying optimal routes in logistics or strategies for high-speed trading. Here, we present a novel method that integrates particle swarm optimization (PSO), a highly effective and widely used algorithm inspired by the collective behavior of bird flocks searching for food, with the physical principle of conserving energy and damping in harmonic oscillators. This physics-based approach allows smoother convergence throughout the optimization process and wider tunability options. We evaluated our method on a standard set of test functions and demonstrated that, in most cases, it outperforms its natural competitors, including the original PSO, as well as commonly used optimization methods such as COBYLA and Differential Evolution.

## 1 Introduction

In general, all optimization methods involve tuning a set of parameters (parameter space) to minimize a function that depends on these parameters (cost function). In most cases, the parameter space is too large to be exhaustively searched, and the most efficient techniques combine stochastic elements (randomness in initial settings and decision-making during the optimization process) with well-designed deterministic processes (following a local gradient or direction based on historical results). Consequently, there is no universal "best" optimization method; instead, different methods and their configurations are more or less effective depending on the specific problem and their success can only be evaluated in a statistical manner.

Metaheuristic optimization techniques are a popular way to perform unconstrained minimization of complicated functions. These techniques are often inspired by natural phenomena, animal behaviors, or evolutionary concepts, making them easy to learn, implement, and hybridize. In addition, they are flexible and can be applied to various problems without altering their structure, treating problems as black boxes where only inputs and outputs matter. Unlike gradient-based approaches, metaheuristics optimize stochastically without a need to

Research and Innovation Authority. The funders did not play any role in study design, data collection and analysis, decision to publish, or preparation of the manuscript. URL: https://vaia.gov.sk/en/2023/08/fellowships-for-excellent-researchers-r2-r4/ https://www.minedu.sk/vedecka-grantova-agentura-msvvam-sr-a-sav-vega/.

**Competing interests:** The authors have declared that no competing interests exist.

differentiate, making them suitable for complex problems. Finally, their stochastic nature also helps avoid local optima, making them effective for challenging real-world problems with complex search spaces.

Despite a large number of different metaheuristic methods published over the years, there is still ongoing research in this area, with current approaches being enhanced and new metaheuristics being proposed frequently. As stated in the No Free Lunch (NFL) theorem [1], no single metaheuristic is best for all optimization problems. An algorithm might perform well on one set of problems but poorly on another. This drives ongoing improvements and the development of new metaheuristics, motivating our efforts to create a new one.

The more standard metaheuristics aiming to optimize a single cost function can be classified into single-solution-based and population-based methods. Single-solution methods, such as Simulated Annealing [2], start with one candidate solution that is improved iteratively. In contrast, population-based methods, such as particle swarm optimization [3] (PSO), begin with multiple solutions that are enhanced over iterations. The advantages of population-based methods include information sharing among solutions, leading to sudden jumps to promising areas, mutual assistance in avoiding local optima, and generally greater exploration compared to single-solution algorithms.

Population-based meta-heuristics algorithms can further be classified into three main branches: evolutionary, physics-based, and swarm intelligence algorithms. Evolutionary algorithms, inspired by natural evolution, optimize by evolving an initial population of random solutions. The most popular algorithm in this class is the Genetic Algorithm (GA) [4], which simulates Darwinian concepts. Each new population is formed by combining and mutating individuals from the previous generation, with the best individuals more likely to contribute, ensuring gradual improvement. Other evolutionary algorithms include Differential Evolution (DE) [5], Evolutionary Programming (EP) [6], Evolution Strategy (ES) [7], Genetic Programming (GP) [8], and Biogeography-Based Optimizer (BBO) [9].

A distinct subclass of population-based evolutionary algorithms is formed by Quality-Diversity (QD) optimization methods, that pursue a single objective, yet deliberately retain multiple solutions distinguished by designer-chosen behavioural traits–the feature space. Every new solution is placed in a cell of this space: if the cell is empty, the solution is kept for being novel; if the cell already contains another, the algorithm keeps whichever of the two has the better objective score. By repeating this simple rule, QD methods build an archive that gradually fills with one high-quality representative for every region of behaviour, giving a rich set of diverse, top-performing options instead of just a single "best" answer. Prominent examples include MAP-Elites [10] and Novelty Search with Local Competition (NSLC) [11], all nicely summarized in [12], demonstrating their versatility in tackling complex, multimodal optimization tasks.

The second main branch of metaheuristics is physics-based techniques which mimic physical rules. One such popular algorithm is Big-Bang Big-Crunch (BBBC) [13], which is inspired by cosmological theory of universe's expansion and contraction. In the Big Bang phase, solutions are randomly dispersed to explore the search space, while in the Big Crunch phase, they are collapsed toward a central point to refine the solution iteratively. Similarly, the Gravitational Search Algorithm (GSA) [14] is based on Newtonian gravity, where candidate solutions are treated as masses that interact through gravitational forces. Over iterations, stronger masses (better solutions) attract weaker ones, leading to convergence toward optimal solutions while balancing exploration and exploitation. Other notable methods include Charged System Search (CSS) [15], Central Force Optimization (CFO) [16], Artificial Chemical Reaction Optimization Algorithm (ACROA) [17], Black Hole (BH) algorithm [18], Ray Optimization (RO) algorithm [19], Small-World Optimization Algorithm (SWOA) [20],

Galaxy-based Search Algorithm (GbSA) [21], and Curved Space Optimization (CSO) [22]. These algorithms use a random set of search agents that move and communicate according to physical rules such as gravitational force, ray casting, electromagnetic force, and inertia. An insightful review of physics-based optimization methods can be found here [23,24]. These physics-inspired algorithms, despite their theoretical origins in fundamental physical laws, have demonstrated remarkable versatility across diverse application domains. They have been successfully deployed in optimizing power flow systems [25], enhancing data clustering methodologies [26] and advancing DNA coding techniques [27,28], among numerous other fields.

The third subclass of metaheuristics is Swarm Intelligence (SI) methods, which mimic the social behavior of groups in nature. Similarly to physics-based algorithms, these use search agents that navigate through collective intelligence. The most popular SI technique is Particle Swarm Optimization (PSO), proposed by Kennedy and Eberhart [3], inspired by bird flocking behavior. PSO employs multiple particles that move on the basis of their own best positions and the best position found by the swarm. Other algorithms in this class are Ant Colony Optimization (ACO) [29], Artificial Bee Colony (ABC) [30], Bat-inspired Algorithm (BA) [31], Marriage in Honey Bees Optimization Algorithm (MHBO) [32], Artificial Fish-Swarm Algorithm (AFSA) [33], Termite Algorithm (TA) [34], Wasp Swarm Algorithm (WSA) [35], Monkey Search (MS) [36], Bee Collecting Pollen Algorithm (BCPA) [37], Cuckoo Search (CS) [38], Dolphin Partner Optimization (DPO) [39], Firefly Algorithm (FA) [40], Bird Mating Optimizer (BMO) [41], Krill Herd (KH) [42], Fruit fly Optimization Algorithm (FOA) [43] and Grey Wolf Optimizer [44]. For more on this topic, see also the comprehensive review on Swarm algorithms [45,46].

Among this pool of methods, Particle Swarm Optimization (PSO) stands out as a significant and relevant algorithm for several reasons such as its simplicity and ease of implementation, as well as outstanding results in test cases and real deployment. Unlike Genetic Algorithms (GA) and Ant Colony Optimization (ACO), PSO requires fewer parameters to adjust, resulting in a reduced computational burden [47–49]. PSO also benefits from its internal "memory" capabilities; it leverages previously known best positions to enhance search efficiency, unlike GAs. Its scalability is another key feature, as PSO can effectively handle complex, high-dimensional optimization problems. In terms of global optimization, PSO demonstrates the ability to avoid multiple local minima, allowing it to navigate rugged landscapes and identify the global minimum. Additionally, PSO's flexibility enables it to be easily hybridized with other optimization techniques, further improving its results [50–52].

As one could expect, despite the numerous benefits mentioned above, PSO also has some drawbacks. One of the main issues they face is the occasional uncontrolled movement of some of the birds in the flock, which can lead to poor convergence. More specifically, even if the optimal points determined by the birds so far are confined to a small region, the birds can gain very high velocities, resulting in an expansion of the search space into irrelevant, large regions. On other occasions, birds lose their velocity very quickly and then proceed to get stuck on a point despite the optimization needing to continue. This balance between these two behaviors is very tight and effectively prohibits any tunability in the convergence process.

A natural approach to address these issues is to get under control the reduction as well as the possible increase of the velocity of the birds. However, as we show in the next section, this is by far not trivial and the existing set of settings of the hyperparameters of PSO can hardly be used. This is why we introduce a new population-based metaheuristic method that introduces the concept of energy (a combination of velocity and distance from the optimal positions known so far) into PSO. By incorporating harmonic motion, the HOPSO algorithm enhances the exploration and exploitation capabilities by providing a more finely tuned

search. Instead of making abrupt, random jumps, each particle oscillates smoothly around a strategic attractor—a balance between its personal and the swarm's best positions. Therefore, not only does this algorithm completely prevent velocity explosions and abrupt death of the birds while keeping all existing PSO features intact, it also allows greater control compared to the original PSO method. This makes the HOPSO algorithm especially adept at avoiding local optima and achieving more reliable optimization outcomes. While the main design inspiration is based on the PSO algorithm, this method is better classified as a physics-based metaheuristic because the movement of the particle population is governed by the physics of harmonic oscillators – hence we name the method as harmonic oscillator-based particle swarm optimization (HOPSO).

The paper is organized as follows: In Section II, we review the existing PSO method and its downsides. In Section III, we introduce our method HOPSO. The results of the HOPSO method are described in Section IV, showcasing its capabilities in 12 different test functions, while Section V concludes our findings.

## 2 Introduction to PSO and its downsides

As briefly described before, the Particle Swarm Optimization (PSO) is a population-based metaheuristic optimization method that works by simulating the social behavior of a flock of birds or a school of fish. In these social systems, the movement of the swarm species was observed to be a form of optimization in their search for food.

Modeling after this swarm behavior, the PSO was developed such that each particle represents a potential solution, and it modifies its position in the search space according to the individual experience (a "cognitive" term) and the group experience (a "social" term). PSO frequently outperforms other algorithms on problems with a smooth landscape, where the optimization process would benefit from a more exploratory approach that can be realized by the collective behavior of the swarm – this is particularly important in the newly developing field of Variational Quantum Algorithms [53], [54].

To briefly describe the operation of PSO, the algorithm begins with a population of $N$ particles, each representing a candidate solution in the $d$ dimensional search space of the optimization problem under consideration. The positions of these particles are randomly initialized within predefined search space boundaries and move according to specific update equations in discrete time steps, i.e. iterations. Specifically, the determination of the position in the subsequent iteration is dependent on the current position with an additional velocity vector that drives the particle to a new, and ideally better, position in the search space. This velocity vector for the next iteration incorporates three crucial components: inertia (previous velocity), cognitive component, and social component. The cognitive component represents the element-wise difference between the personal best position vector and its current position vector, while the social component represents the element-wise difference between the global (swarm) best position vector and the particle's current position. Each of these is then scaled by the 'cognitive' and 'social' coefficients, denoted as $c_1$ and $c_2$, respectively. These coefficients represent the relative importance assigned to the particle's personal best position (stored in its individual memory) and the swarm's global best position when calculating the velocity for the next iteration, thereby determining the relative importance of the particle's personal experience versus the swarm's collective knowledge in calculating the next velocity and position. Moreover, each of these terms is adjusted by a unique random factor, which introduces stochasticity into the search process. This movement is illustrated in Fig 1.

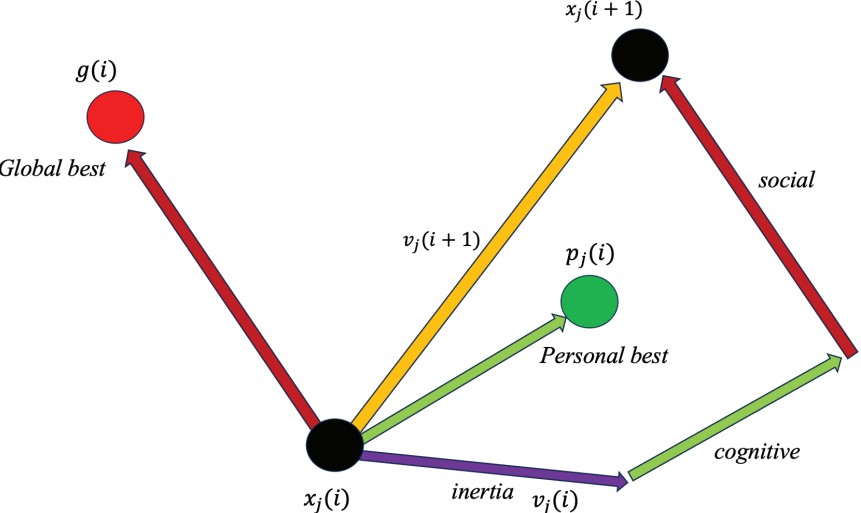

**Fig 1. An example of the movement of a particle in a two-dimensional space based on the PSO algorithm in a single iteration.** An inertia term given by velocity that drives the particle in some direction (violet arrow), a memory term ($p_j$) that influences the particle's trajectory based on its best known position (green arrow), and a global-best cooperation term ($g$) that reflects the best result amongst the entire swarm (red) constitute the particle's projected movement (yellow arrow). The $i$ indicates iteration, while $j$ indicates particle number.

To allow convergence of the whole system, a damping factor in the form of a *constrictor factor $\chi$* was introduced in [55]. The update equations of this variation of the PSO algorithm are introduced as follows with a description of its variables in Table 1.

**Velocity Update:**

$$v_{j,d}(i+1) = \chi\big(v_{j,d}(i) + c_1 r_1\big(p_{j,d} - x_{j,d}(i)\big) \\ + c_2 r_2\big(g_d - x_{j,d}(i)\big)\big) \tag{1}$$

**Position Update:**

$$x_{j,d}(i+1) = x_{j,d}(i) + v_{j,d}(i+1) \tag{2}$$

**Table 1. Description of PSO parameters.**

| Parameter | Description |
|---|---|
| $v_{j,d}(i)$ | Velocity of particle $j$ in dimension $d$ at iteration $i$ |
| $x_{j,d}(i)$ | Position of particle $j$ in dimension $d$ at iteration $i$ |
| $\chi$ | Constrictor factor (damping) |
| $c_1$ | Cognitive coefficient, attraction towards the particle's best known position |
| $c_2$ | Social coefficient, attraction towards the swarm's best known position |
| $r_1, r_2$ | Random values uniformly distributed in the range [0,1] |
| $p_{j,d}$ | $d$-th dimension of the best known position of particle $j$ |
| $g_d$ | $d$-th dimension of the global best known position |

## 2.1 Convergence analysis without randomness

Balancing exploration capabilities and reasonable convergence is the main challenge in all variants of PSO. A low constriction factor causes particles to stop moving too quickly, while high values (leading to low damping) cause particles to spread into large regions far from any optima. Therefore, a specific form of the constrictor factor, $\chi$, has been derived from the stability analysis of the PSO system to address these concerns about the values of the velocity term.

The velocity update equation can be viewed as a second-order difference equation, which occurs when removing random variables from the PSO equation and is then examined using roots of its characteristic equation. These roots, $\lambda$'s, determine the behavior of the velocity over time and thereby, for the system to be stable (converge), the absolute value of the roots must be less than 1.

The key parameters determining the value of $\lambda$'s are $c_1$ and $c_2$, entering the equations as their sum $\varphi = c_1 + c_2$, as well as the constrictor factor $\chi$. Larger values of $\varphi$ generally cause the need for the constrictor factor to dampen the velocities more significantly to prevent runaway divergence, therefore allowing for better and more controlled convergence behavior. Clerc and Kennedy proposed the mathematically designed form of $\chi$ as seen below in Eq (3)

$$\chi = \frac{2}{\left| 2 - \varphi - \sqrt{\varphi^2 - 4\varphi} \right|} \tag{3}$$

$$\varphi = c_1 + c_2. \tag{4}$$

This fully determines the form of $\chi$ in no-randomness scenario.

Empirical studies suggest that $c_1 = c_2 = 2.05$ [56] is a particularly effective set of parameters because it balances exploration and exploitation: the swarm maintains enough "pull" toward both personal and global best positions to explore the parameter space without overshooting. This combination of parameters produces a constrictor factor $\chi = 0.7298$, derived from Eq (3). This value of $\chi$ helps prevent particles from diverging (velocity "explosions") or stagnating prematurely, thus promoting a smoother convergence and proving to be very effective in many cases.

## 2.2 Velocity explosions

Let us now analyze in detail the potential for velocity explosions in PSO, a phenomenon described by Kennedy and Eberhart [55]. This occurs when particle velocities significantly exceed the characteristic scale of the search space during the optimization process, leading to swarm divergence, as these high-velocity particles continue to traverse the parameter space rapidly, and their ability to effectively locate optima of the cost function becomes severely compromised. This behavior thereby undermines the balance between exploration and exploitation that is crucial for the PSO's effectiveness.

In order to understand why these velocity explosions occur, let us briefly analyze the time evolution of a single bird in our swarm. For simplicity, let us consider only one dimension, as each of the dimensions behaves independently until a new optimum is identified. In this case, the position update Eq (2) and the velocity update Eq (1) can be represented with a matrix $M$ acting on the vector with position and velocity coordinates, which thereby provides a compact way to describe the coupled dynamics of both velocity and position written as presented in Eq (5)

$$\vec{P}_{t+1} = M\vec{P}_t, \quad \vec{P}_t = (v_t, y_t), \tag{5}$$

where

$$y_t = \frac{\varphi_1 g + \varphi_2 p}{\varphi_1 + \varphi_2} - x_t, \tag{6}$$

with $\varphi_1 = r_1 c_1$ and $\varphi_2 = c_2 r_2$ and the dynamical matrix $M$ governing the time evolution defined as

$$M = \begin{pmatrix} \chi & \chi\varphi \\ -\chi & 1 - \chi\varphi \end{pmatrix}, \tag{7}$$

where $\varphi = \varphi_1 + \varphi_2$.

Notice that Eq (5) only represents a single iteration step but since the random numbers $r_1$ and $r_2$ are changing in each iteration, to represent the state of the particle after several iterations we need to accumulate the effects of all previous iterations; therefore, the state vector $\vec{P}_t$ of a particle that was initially at position $x_0$ with velocity $v_0$ can be described as a product of transformations

$$\vec{P}_t = \prod_{i=0}^{t} M_i \vec{P}_0, \quad \vec{P}_0 = (v_0, y_0). \tag{8}$$

If no new local or global optimum is found, it is expected that the particle will gradually converge and its velocity will decay to zero. As analyzed in [55], when one keeps the same value of $\chi$ and $\varphi$ throughout the entire simulation (removing the stochastic nature of the method), the sufficient condition for convergence is

$$\max(|\lambda_1|, |\lambda_2|) < 1, \tag{9}$$

where $\lambda_1$ and $\lambda_2$ are the eigenvalues of $M$ given by

$$\lambda_{1,2} = \frac{1}{2}\left(1 + (1 - \varphi)\chi \pm \sqrt{((\varphi - 1)\chi - 1)^2 - 4\chi}\right), \tag{10}$$

as both the eigenvectors are multiplied by $\lambda_1^t$ and $\lambda_2^t$ which both converge to 0.

This is no longer the case when one includes the randomness in the PSO. In this case, the behavior is governed by a product of $t$ different dynamical matrices in Eq (8) . Here, despite the fact that each individual matrix $M$ has the magnitude of both eigenvalues below 1, it is in general not the case for their product, i.e. subsequent multiplication of matrices, with each of their eigenvalue being smaller than one, can form a matrix with a very large eigenvalue, and therefore leading to explosions in velocities.

To analyze the product of different matrices, a more suitable approach is to look into the singular values given by

$$\sigma_{1,2}^2 = \frac{1}{2}\Big(2\chi^2\big((c_1 r_1 + c_2 r_2)^2 + 1\big) - 2(c_1 r_1 + c_2 r_2)\chi + 1$$
$$\pm \sqrt{\big(2\chi^2\big((c_1 r_1 + c_2 r_2)\big)^2 + 1\big) - 2(c_1 r_1 + c_2 r_2)\chi + 1\big)^2 - 4\chi^2}\Big). \tag{11}$$

Singular values express the limits of the maximum possible prolongation or contraction of a vector that is multiplied by a given matrix. To have the length of the vector under control, one optimally needs to have both singular values similar, and for smooth convergence, both of them must be just slightly smaller than one.

Using a simplification of $c_1 = c_2 = 2.05 = c$ along with definition of $r = r_1 + r_2$ (as in fact only the sum of random number enters the analysis), Eq (11) becomes

$$\sigma_{1,2}^2 = \frac{1}{2}\left(2\chi^2\left((cr)^2 + 1\right) - 2cr\chi + 1 \pm \sqrt{\left(2\chi^2(cr)^2 + 1\right) - 2cr\chi + 1\right)^2 - 4\chi^2}\right). \tag{12}$$

The singular values for matrix (7) are shown in Fig 2 – note that the sum of random numbers runs from 0 to 2.

One can see that in the region of small sum of random numbers, both singular values are close to one, so one can expect some kind of reasonably small convergence. However, for large sum of random numbers this is not the case – one singular values almost diminishes and the other one reaches almost the value of 4, thus one can, depending on the starting position and exact combination of random numbers obtain both very fast dying out, as well as velocity and position explosions.

## 2.3 Influence of randomness

Our observations from running simulations have shown that the explosions result from a specific series of random numbers, where large and small numbers have followed each other regularly. This is because the specific form of the matrix $M$ causes rotation in the governing vector (large velocity in one step induces large distance in the next one and vice versa), leading to the application of the second singular value in the next step. So a simple series of large (or small) random numbers will rather cause oscillations that are damped by a suitable constriction factor. However, if repeatedly a large random number is applied when the larger singular value is active along with a small random number combined with smaller singular value, the resulting behavior is divergent, On contrary, repeated combination of large random number when small singular value is active with subsequent small random number for small singular value will cause a very quick death - in such a case, the particle can only be revived by finding a new global maximum.

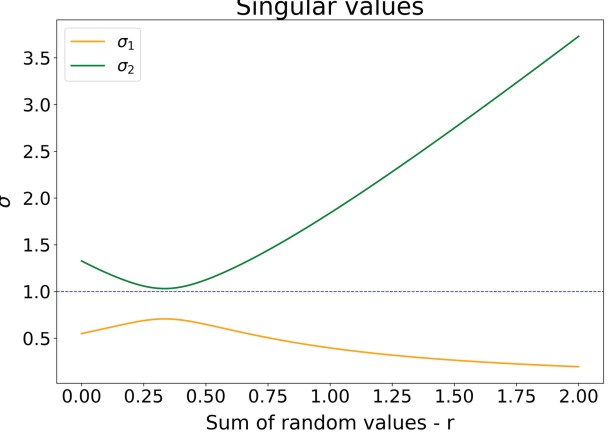

**Fig 2. The effect of randomness in the singular values of the dynamic matrix $M$, with $c_1 = c_2 = 2.05$ and for $\chi = 0.729$.** For large $r$, the difference between the singular values is large, increasing the probability of velocity explosions or death.

It has been also been observed in the past and confirmed in our data that using a single random number $r$ rather then using two independent random numbers $r_1$ and $r_2$ makes the probability of velocity explosions much higher. From our analysis the reason became clear – with two independent random numbers the probability of their sum being particularly small or large is much smaller than when taking one number instead, making the explosions less probable. This can also be seen on a simple example – probability of single random number with flat distribution from 0 to 1 being larger than 0.9 is 10%, whereas an average of two random numbers taken from the same distribution is larger than 0.9 with probability of about 1%.

It is clear that achieving a smooth and controlled convergence of the method with its existing governing equation would require a very detailed control of the applied randomness. However, this would compromise the stochastic nature of the method: the more restrictions that are applied, the higher is the probability of not reaching the global maximum. Thus, we suggest reformulating the idea of PSO in a more physical framework, introducing the concept of energy, its conservation, and loss via damping, together with possible energy boosts connected with finding a new global or local optimum.

## 3 Harmonic oscillator based particle swarm optimization – HOPSO

The general framework of energy consists of two basic types: kinetic energy, defined by the movement of particles, and potential energy, defined in most cases by the position of particles and expressing the potential to gain (or lose) kinetic energy by changing its position. Potential energy can only be associated by conservative forces, such as gravity, electromagnetic forces, or elastic forces. If only these kind of forces act in a system, the total energy is conserved - it can only be transformed between its kinetic and potential type, but it can never emerge or disappear.

In systems that are not described at the micro-level of atoms and molecules, such as mechanical and thermodynamical systems, forces that cannot be associated with a potential energy can appear. These forces are called non-conservative and friction forces are typical examples of them. They lead to loss of energy (or, more precisely, to dissipation into heat or other types of energy), where the sum of potential and kinetic energy is gradually decreased.

Here we introduce the concept of energy and its conservation combined with a tunable dissipation on top of the PSO approach. While there exist optimization approaches that are based on gravity [14], [57], which are mathematically very similar to the model of charges, we consider it as not a very promising approach. Firstly, the energy diverges to negative infinity while reaching the center of gravity (the current optimum), again leading to explosions of velocities. Secondly, for long distances the attractive force practically diminishes, allowing the particles to fly very far from the attractor.

For such reasons, we were inspired to use elastic forces, e.g. forces that are linear with the distance between the attractor and the position of the particle, and can be associated with an ideal spring. The potential energy therefore follows Hooke's Law—growing with the square of the distance between the particle and the attractor, meanwhile the kinetic energy (with assumed unit mass) reflects the velocity of the particle. This concept is optimal because for small distances the potential energy converges to zero, while for large distances the energy grows quadratically to infinity, essentially bounding the particle to a well-defined region.

One challenge is that when a single spring is used across multiple dimensions, the total energy depends in a complex non-trivial way on the position in all dimensions. To simplify the process, we propose a model in which an independent virtual spring is assigned to each

dimension, attracting only the particle within that dimension. This approach maintains the principle of energy conservation while enabling faster and simpler calculations. It also allows for independent adjustment of constraints in different dimensions, as the energy is now decoupled for each parameter.

Now, the movement of particles, if modeled in continuous time, would reflect swinging on a spring with a center defined by an attractor (a suitable combination of social and cognitive term) independently in each single dimension. The period of these oscillations is defined by a combination of a virtual mass of that particle and the stiffness of the spring, each of which can be chosen arbitrarily. This movement has a simple analytical solution, the harmonic motion.

In the optimization process, there is no need to model the entire motion. Instead, snapshots are taken at different moments during the harmonic oscillations. Randomness starts to influence the process in HOPSO at this stage. Rather than altering the potential (by adjusting the stiffness of the spring and the mass of the particle), we instead allowed the particles to oscillate harmonically with a unique frequency and observed them as snapshots at different time intervals.

In contrast to the original PSO where the damping influences multiple aspects including the social and cognitive terms through the constrictor factor, here the damping constant solely governs energy dissipation, which directly relates to the searching ability of the particle. In the results section, we will show how to use this parameter, indicating that not only does this parameter offer more tunability in governing convergence via this physically inspired model, but also that this damping parameter is an easily controlled independent parameter (i.e., can be classified as a "free parameter"). This flexibility offers a significant advantage for optimization problems and is one of the crucial advantages of the HOPSO algorithm over not only the standard PSO method, but other non-gradient methods as well, presented in our results section.

## 3.1 Formal definitions

More formally, each particle's position in each dimension is defined as the solution of a damped harmonic oscillator. The solution is given by the following equation:

$$x(t) = A_0 e^{-\lambda t} \cos(\omega t + \theta) + x_0, \tag{13}$$

where $x(t), A_0, \lambda, \omega$ and $\theta$ represent the position of the particle at time $t$, initial amplitude, damping factor, angular frequency and initial phase of the oscillation, respectively, whereas the $x_0$ is the position of the attractor.

The velocity of the particle at any given time $t$ can be obtained by differentiating Eq (13)

$$v(t) = -\omega(A_0 e^{-\lambda t} \sin(\omega t + \theta)) - \lambda(x(t) - x_0). \tag{14}$$

The measurement time is chosen randomly within the interval $[0, t_{ul}]$ for each particle and in each dimension, where $t_{ul}$ is the upper limit of the time sampling range. For small values of $t_{ul}$ the sampling range does depend on a particular choice of this time. However for $t_{ul} > \pi$ the range covers the whole oscillation space of the particle and for any multiples of any multiples of $\pi$ the coverage is the same. For simplicity, we use $t_{ul} = 2\pi$ to cover the whole period of oscillations. The iterative change in parameter $t$ is then defined as

$$t_{i+1} = t_i + \text{rand}[0, t_{ul}], \tag{15}$$

where the parameter $i$ represents the index of iteration.

The optimization begins by initializing the particles at random positions along with random velocities in the parameter landscape. The initial positions are noted as the initial personal best positions. The global best position is noted as the best position out of all the personal best positions of all the particles. This global best position corresponds to the lowest value of the cost function within the swarm.

Each particle then oscillates about an attractor independently in each dimension. This attractor can be calculated as

$$a_{j,d} = \frac{c_1 p_{j,d} + c_2 g_d}{c_1 + c_2},$$

(16)

where $a_{j,d}, p_{j,d}, g_d$ represent, respectively, the position of the attractor, personal best position of the particle and the global best position for the $j^{th}$ particle in $d^{th}$ dimension. The $c_1$ and $c_2$ terms represent the weights of attraction towards the personal-best and global-best positions, respectively. Typically, the values $c_1$ and $c_2$ are set equal, so the attractor lies equidistant between the personal best position and the global best position.

We solve Eq (13) and Eq (14) to obtain the initial amplitudes $A_0$ for each particle and in each dimension by choosing the initial time as $t = 0$:

$$A_0 = \sqrt{(x(0) - a)^2 + \frac{(v(0) + \lambda(x(0) - a))^2}{\omega^2}}.$$

(17)

Once the initial oscillation amplitude $A_0$ is determined, the initial phase of the oscillation $\theta$ can be calculated as

$$\theta = \arccos \frac{x(0) - a_{j,d}}{A_0}.$$

(18)

We then let the particles oscillate in time, with the amplitude of the oscillation decaying as $A_0 e^{-\lambda t}$. For every iteration, we stop the clock at a random time, calculate the values of the cost function for all the particle positions.

If there is a change in $p_j$, then the attractor for the $j^{th}$ particle is recalculated using Eq (16) while the amplitudes and phase values are recalculated by resetting the time for that particular particle as zero using Eq (17) & Eq (18). If, instead, there is a change in $g$, then all the attractors are changed accordingly using Eq (16). The amplitudes and phases are again recalculated using Eq (17) & Eq (18) by resetting the time as zero for all particles.

## 3.2 Energy rewards for finding a new optimum

In some cases, this procedure might lead to a significant loss of energy for the particle that found the new best position. This can be seen in an example where the best position is found at the boundary of the oscillation region where the velocity of the particle is close to zero. If that position is the new attractor, the potential energy of the string will be zero as well, as the length of the spring will be zero. This effectively freezes the particle until a new global best position is found. This is naturally not a desirable situation, as that particle is "punished" for being successful. To deal with this issue, we postulate a condition that the newly calculated amplitude is never smaller than the previous one – more generally, finding a new best position can never lead to decrease of the energy in the system. In other words, the particle receives an energy reward for finding a new optimum that exactly compensates the loss in potential energy associated with change of the spring's origin.

## 3.3 No damping for distant particles

Given enough time without finding a new global best, there is a possibility that the amplitude of particles whose personal best is far away from the global best becomes so small that the particle fails to effectively search the space between them, causing it to virtually become stuck in the middle, which might be a very unfavorable region. Numerical simulations did show that this basically leads to an effective loss of this particle, as it will almost never contribute with new results. A naive approach to solve this problem is by reducing the damping parameter $\lambda$, however the drawbacks of this is that this may also reduce the convergence in the desired cases.

We therefore included a different approach to resolve this problem, namely, a special rule is introduced to limit the amplitude from below to some threshold amplitude $A_{th}$ and prevent it from further damping. This threshold amplitude is naturally chosen as a reasonable multiple of the distance between the global and personal best positions to allow for searching in a region encompassing both of them. Thus, we define

$$A_{th} = \frac{|p_{j,d} - g_d|}{2}*m,\tag{19}$$

where $m$ is a parameter greater than one. A pictorial representation of this thresholding is shown in the Fig 3.

A pseudocode formalizing all the procedures described above is shown in Algorithm 1.

## 4 Results

We now demonstrate the performance of our algorithm on a large set of commonly used test functions. These functions are listed in Table 2.

These particular functions were chosen for their different and diverse properties, making them ideal for testing optimization algorithms [58], [59] and are a common standard choice. The Ackley and Rastrigin functions have many local minima. Similarly, the complex landscape of the Levy function challenges algorithms to avoid local minima. The Sphere function

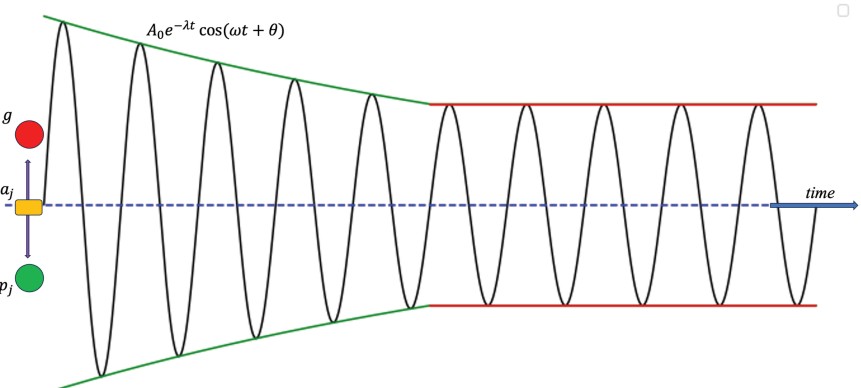

**Fig 3. Visualization of HOPSO in one-dimension.** In one-dimension, the particle $j$ oscillates about the attractor $a_j$ which is set half-way between its personal best ($p_j$) and the swarm's global best ($g$) based on the weighted average equation (16). The damping is switched off when the amplitude decreases, in the depicted case to approximately twice ($m = 2.05$) the distance between the attractor and one of the best positions.

**Algorithm 1.    Harmonic oscillator based particle swarm optimization (HOPSO).**

**Input:** Problem dimensions, Objective function, Algorithm parameters
**Output:** Candidate Optimal solution
Set constants $c_1$, $c_2$, $\lambda$, m for attraction weights and damping and minimal
 amplitude;
Initialize particles with random positions $x_{j,d}$ and velocities $v_{j,d}$;
Set initial personal best positions $p_i$ by starting positions for each
 particle;
Choose initial global best position g;
Calculate position of attractors using Equation 16;
Calculate initial amplitude using Equation 17;
Calculate initial phase using Equation 18;
**while** *iteration < max_iterations* **do**
 **foreach** *particle* **do**
 **foreach** *dimension* **do**
 Calculate dynamic amplitude as $A_0 e^{-\lambda t}$ while accounting for the
 threshold amplitude using Equation 19;
 Update position using Equation 13;
 Update velocity using Equation 14;

 **foreach** *particle* **do**
 Calculate Cost function from positions;
 **if** *Cost_function($x_{j,d}(t)$) < Cost_function($p_j$)* **then**
 Update $p_j$, best value, time, attractors, amplitude, phase;

 **if** *personal best value < global best value* **then**
 Update global best value g;
 **foreach** *particle and dimension* **do**
 Reset time, recalculate attractors, amplitude, phase;

 iteration ← iteration + 1;

is a simple, unimodal, bowl-shaped function, while the Beale function is multimodal with sharp peaks at the corners of the input domain. The Goldstein-Price function is highly multimodal and complex, while the Schwefel function, known for its large search space, presents numerous traps. The Rosenbrock function's narrow, curved valley tests precision, while the Drop-Wave function features steep drops and peaks. The Cross-in-Tray function and the Michalewicz function, with their deep valleys and sharp peaks, also make it extremely challenging for optimization algorithms to find the optimum and are therefore suitable choices as well. Each function was chosen to test uniquely different aspects of the optimization algorithm that we present in this text.

The performance of HOPSO will be compared with that of the standard optimization methods PSO, COBYLA [60] and Differential Evolution (DE) [61]. The code for the HOPSO algorithm was written in Python and can be found here [62].

Constrained Optimization by Linear Approximation (COBYLA) is a simplex-based gradient-free optimization method. It was first introduced by Michael J.D. Powell in 1994 [60]. To briefly describe the algorithm, it operates by creating a simplex, a polytope of $n + 1$ vertices for an $n$-dimensional space, and using the values of the objective function at the vertices of this simplex, it approximates the objective function along with linear constraints, solving linear programming problems within a trust region. The simplex and the trust region are adjusted iteratively until the convergence is obtained. For more details, refer to Powell's original paper [60].

Differential evolution (DE) is another metaheuristic algorithm like PSO. DE is an optimization technique that begins with a set of possible solutions and gradually refines them. DE was first given by Rainer Storn and Kenneth Price in 1997 [61]. To briefly describe DE, a

**Table 2. Commonly used test functions for optimization methods**

| Name | Functional Form | Modality | Search Domain | $F_{min}$ | Dimension |
|---|---|---|---|---|---|
| Ackley | $-a\exp\left(-b\sqrt{\frac{1}{d}\sum_{i=1}^{d}x_i^2}\right)$ $-\exp\left(\frac{1}{d}\sum_{i=1}^{d}\cos(cx_i)\right)+a+\exp(1)$ | Multimodal | [-32.76,32.76] | 0 | 10 |
| Beale | $(1.5-x_1+x_1x_2)^2+(2.25-x_1+x_1x_2^2)$ $+(2.625-x_1+x_1x_2^3)$ | Multimodal | [-5,5] | 0 | 2 |
| Cross-in-Tray | $-0.0001[\|\sin(x_1)\sin(x_2)$ $\exp(\|100-\frac{\sqrt{x_1^2+x_2^2}}{\pi}\|)\|+1]^{0.1}$ | Multimodal | [-10,10] | -2.06261 | 2 |
| Drop-Wave | $-\frac{1+\cos(12\sqrt{x_1^2+x_2^2})}{0.5(x_1^2+x_2^2)+2}$ | Multimodal | [-5.12,5.12] | -1 | 2 |
| Goldstein-Price | $[1+(x_1+x_2+1)^2(19-14x_1+3x_1^2-14x_2$ $+6x_1x_2+3x_2^2)]\cdot[30+(2x_1-3x_2)^2(18$ $-32x_1+12x_1^2+48x_2-36x_1x_2+27x_2^2)]$ | Multimodal | [-2,2] | 3 | 2 |
| Griewank | $\frac{1}{4000}\sum_{i=1}^{d}x_i^2-\prod_{i=1}^{d}\cos\left(\frac{x_i}{\sqrt{i}}\right)+1$ | Multimodal | [-600,600] | 0 | 10 |
| Levy | $\sin^2(\pi w_1)+\sum_{i=1}^{d-1}(w_i-1)^2$ $[1+10\sin^2(\pi w_i+1)]$ $+(w_d-1)^2[1+\sin^2(2\pi w_d)]$ | Multimodal | [-10,10] | 0 | 10 |
| Michalewicz | $-\sum_{i=1}^{d}\sin(x_i)\left[\sin\left(\frac{ix_i^2}{\pi}\right)\right]^{2m}$ | Multimodal | [0,$\pi$] | -4.687 | 5 |
| Rastrigin | $10d+\sum_{i=1}^{d}[x_i^2-10\cos(2\pi x_i)]$ | Multimodal | [-5.12,5.12] | 0 | 10 |
| Rosenbrock | $\sum_{i=1}^{d-1}[100(x_{i+1}-x_i^2)^2+(x_i-1)^2]$ | Unimodal | [-5,10] | 0 | 10 |
| Schwefel | $\sum_{i=1}^{d}[-x_i\sin(\sqrt{\|x_i\|})]$ | Multimodal | [-500,500] | 0 | 10 |
| Sphere | $\sum_{i=1}^{d}x_i^2$ | Unimodal | [-10,10] | 0 | 5 |

population of candidate solutions is initialized randomly. For each candidate, a mutant vector is generated by adding the weighted difference between two randomly selected population vectors to a third vector. This mutant vector is then recombined with the target vector, and a selection process determines whether the new vector replaces the target vector based on a fitness evaluation. This process is repeated iteratively until a stopping criterion is met. Recently, [63] it was shown that DE, although not as popular as PSO, outperforms PSO in many cases, which makes it a natural choice for an optimizer for comparison.

In principle, for a given budget, the same method can achieve different results for different values of the tuning parameters, so we use the commonly used "standard settings" which result in typically good performance for each optimization method. Specifically, here in this study Scipy's optimization module [64] was used without altering its standard settings for each optimizer to implement COBYLA and DE. PSO was run via our own implementation but using the standard parameter settings $\chi = 0.729$ and $c_1$, $c_2$ each being 2.05 [56]. The HOPSO settings are designed to mirror the equivalent parameters in the PSO, ensuring a fair playing field between the optimizers. The HOPSO specific settings are $c_1 = c_2 = \omega = 1$, $t_{ul} = 2\pi$, and $m = 2.05$.

## 4.1 Budget on function evaluations

To compare optimization algorithms, it is essential to define a "budget" based on a specific resource. Using the maximum number of iterations as the budget can be problematic, since different optimizers may use varying numbers of function calls per iteration (e.g. while calculating a gradient), even if they share the same maximum iterations. Therefore, in our case, we select function evaluations as a budget. This budget represents how many times the algorithm can query the objective function to assess its performance. Using function evaluations, we ensure a fair comparison between algorithms regardless of their internal operations. Depending on the complexity of the problem, we chose our budget to be either 1000 or 10000 function evaluations, as listed in Table 3.

## 4.2 Damping in HOPSO

The crucial difference that offers the superior advantage of HOPSO over PSO is the tunability of the damping parameter $\lambda$. Here, we provide a general guideline (or starting point) for selecting $\lambda$ that the user can adjust as needed. As $\lambda$ governs damping, it should naturally be inversely proportional to the budget per particle. For a higher budget of function evaluations per particle, there is less need for damping in the system, since less damping allows for a broader search, and vice versa. As a proportionality constant, we define a scaling factor $s$. The relationship for setting the $\lambda$ based on the budget is as follows

$$\lambda \equiv s \left( \frac{B}{N} \right)^{-1}, \tag{20}$$

where $B$ is the number of function evaluations (i.e. budget), $N$ is the total number of particles, and $s$ is a scaling factor. Extremely large or small values of $s$ are undesirable, as they lead to overly strong or weak damping, respectively.

For the case where $s = 1$, if no updates occur during optimization, the amplitude $A$ reduces to $A/e$ (approximately $0.36A$) over the total time period of $B/N$. With updates and due to the reduced damping in certain cases, the final amplitude might be even higher, resulting in a broad but not very precise search.

**Table 3. Results of the optimization using different optimizers on different test functions. As one can see, the HOPSO method outperforms the standard PSO methods in all cases and in most of the cases its results are the best among all tested methods.**

| Function | Function evaluations | $F_{min}$ | Mean | | | |
|---|---|---|---|---|---|---|
| | | | HOPSO | PSO | COBYLA | DE |
| Ackely | 10000 | 0 | **0.0115** | 1.3824 | 19.410 | 5.6180 |
| Beale | 1000 | 0 | **0.0363** | 0.0635 | 1.0368 | 0.1174 |
| Cross-in-Tray | 10000 | -2.0626 | **-2.0626** | **-2.0626** | -1.7594 | **-2.0626** |
| Drop wave | 10000 | -1 | **-0.9841** | -0.9790 | -0.3781 | -0.9460 |
| Goldstein-Price | 1000 | 3 | **4.080** | 5.4463 | 70.427 | 6.240 |
| Griewank | 10000 | 0 | **0.1033** | 0.1471 | 21.920 | 1.0461 |
| Levy | 10000 | 0 | **0.1749** | 2.0717 | 20.081 | 1.5805 |
| Michalewicz | 10000 | -4.687 | -4.5119 | -4.0539 | -3.0879 | **-4.5187** |
| Rastrigin | 10000 | 0 | 12.458 | 14.298 | 55.091 | **11.770** |
| Rosenbrock | 10000 | 0 | 5.3834 | 7776.2 | 12.612 | **1.1960** |
| Schwefel | 10000 | 0 | 1002.1 | 1083 | 1621.5 | **686.57** |
| Sphere | 1000 | 0 | **0** | **0** | **0** | **0** |

A more aggressive damping can be achieved by choosing $s = 10$, in which case, within the same time period, the amplitude decreases to $0.000045A$ (the decrease is smaller if new optima are found or the damping is not allowed due to distance between the global and personal best) allows a more precise search at a higher risk of falling into a local minimum. In the comparison of our study, we chose 10 for all functions to maintain a fair battlefield. Towards the end of this section, we elaborate on the influence of $\lambda$ in specific cases.

The results are presented in Figs 4–15. In all the boxplots, outliers have been omitted for improved visualization; however, they have been included in the calculation of the means. For Cross-in-Tray, Drop-Wave, Schwefel, Griewank, Levy, Michalewicz and Rosenbrock functions the particles were restricted to only search the domain of interest as mentioned in Table 2.

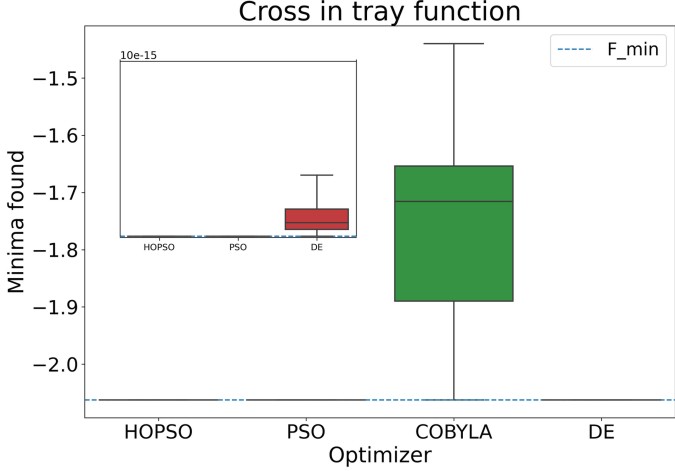

**Fig 4. Performance of the optimizers on cross-in-tray function.** Except for COBYLA, all other optimizers converge to the minima, with HOPSO and PSO performing much better than their competitors.

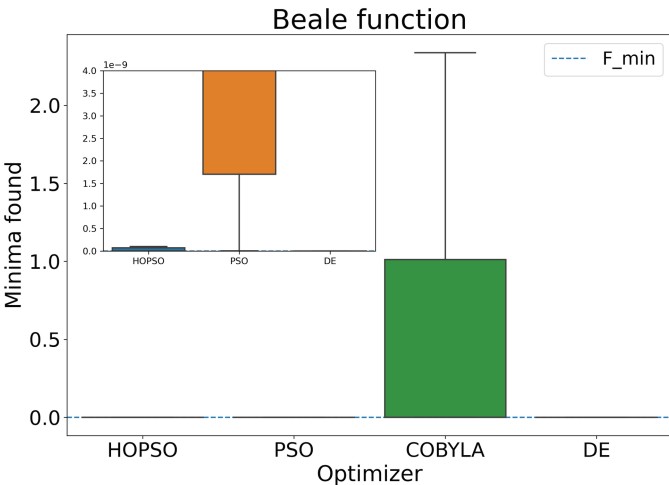

**Fig 5. Performance of the optimizers on Beale function.** All optimizers except for COBYLA converge reasonably well, with PSO and COBYLA failing to reach the precision of HOPSO and DE. DE performs more precisely than HOPSO but fails to consistently optimize the function successfully. This can be seen in Table 3, where for Beale, DE has a higher mean value.

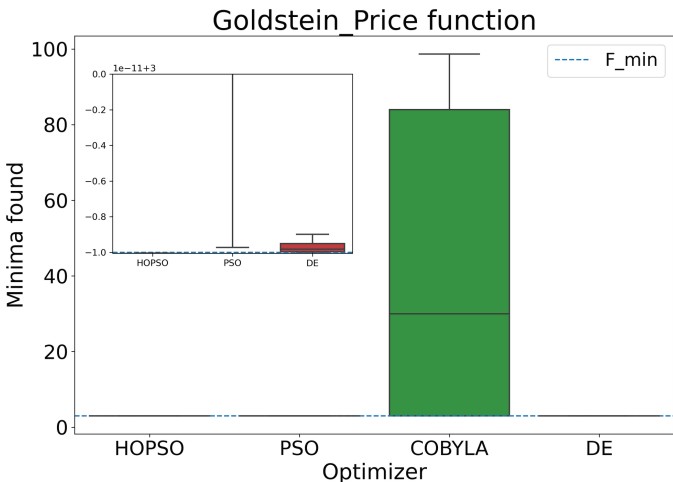

**Fig 6. Performance of the optimizers on Goldstein-Price function.** Here COBYLA performs poorly compared to its competitors, which solve the problem adequately. In terms of accuracy, HOPSO outperforms the other optimization methods.

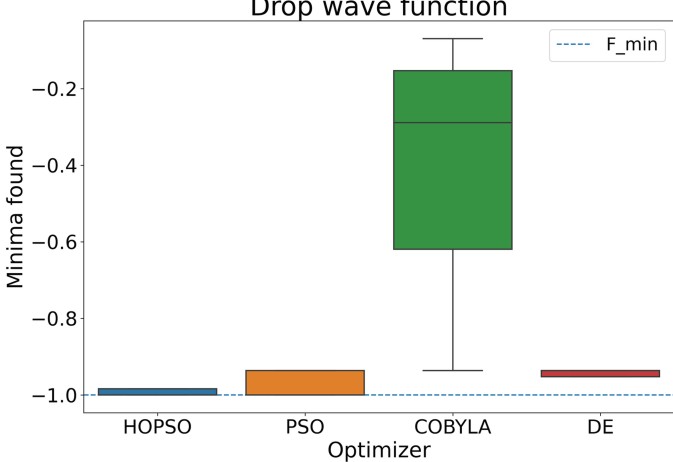

**Fig 7. Performance of the optimizers on Drop-Wave function.** Unlike in previous cases, none of the optimizers converges perfectly within a given budget. Here, both PSO and HOPSO do reach better results than COBYLA and DE, with HOPSO achieving higher accuracy.

As seen in Figs 4–7, for the two-dimensional test functions, HOPSO outperforms the other optimizers in all cases; COBYLA consistently underperforms, while DE ranks second. For the Cross-in-Tray function, all optimizers except COBYLA reach the minimum. Performance trends are similar for the Beale and Goldstein-Price functions. For the drop-wave function, HOPSO is closest to the minimum, followed by PSO, DE, and COBYLA.

In Figs 8–15, which shows test functions with varying dimensions, HOPSO excels at Ackley, Levy, and Griewank functions. Its performance is comparable to DE on the Rastrigin and Michalewicz functions. For the Schwefel function, none of the optimizers reach the minimum (with DE performing best, albeit still poor). In Figs 14–15, for the unimodal test functions, HOPSO, together with PSO and DE, converges to the minimum for the sphere function. For

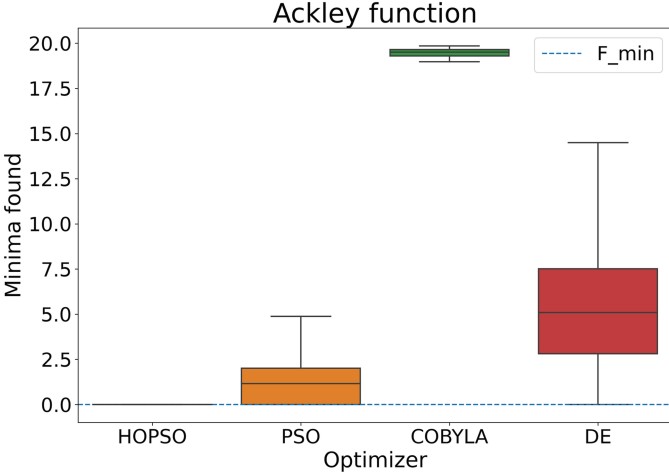

**Fig 8. Performance of the optimizers on Ackley function.** All optimizers except HOPSO fail to converge to the minima, with HOPSO being more precise and more accurate than its competitors.

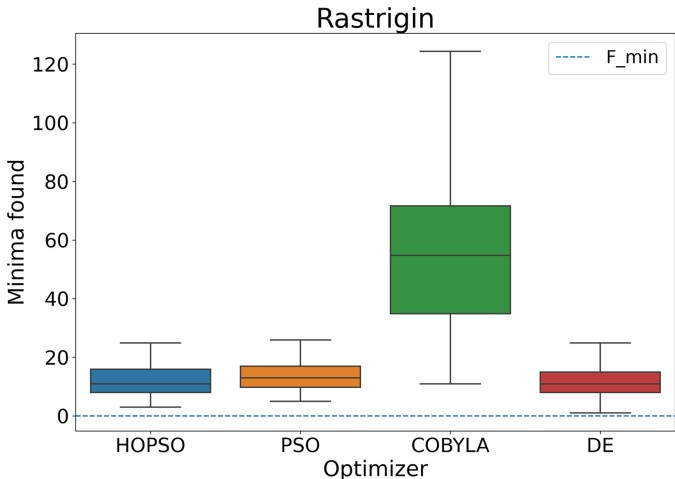

**Fig 9. Performance of the optimizers on Rastrigin function.** All optimizers fail to converge to the minima. However, DE performs the best slightly ahead of HOPSO and PSO which in turn are significantly better than COBYLA.

the Rosenbrock function, HOPSO works significantly better than the original PSO, while it is slightly outperformed by DE and COBYLA.

All in all, HOPSO clearly outperforms the other three optimizers in six cases. In two cases, HOPSO, PSO, and DE perform equally well, while DE performs better in four cases, however, in three of them the results of all optimizers are far away from the actual solution, so one might conclude that all of them did fail.

## 4.3 Tuning of damping in HOPSO

These results for HOPSO were achieved without fine-tuning the hyperparameters, similar to the other optimizers. To investigate the abilities of HOPSO, we tested it for different values

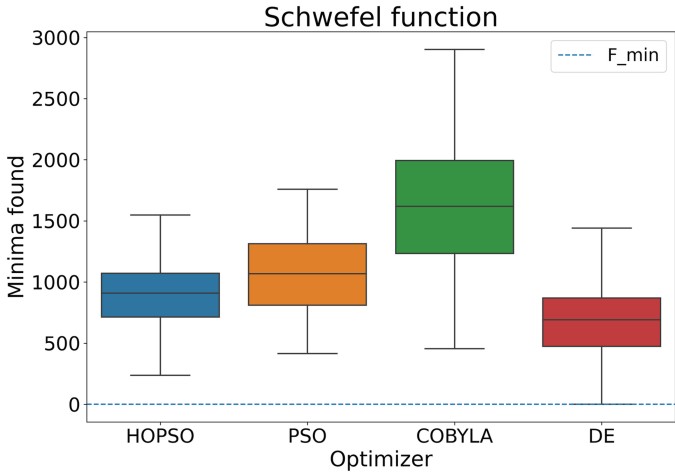

**Fig 10. Performance of the optimizers on Schwefel function.** All optimizers fail to converge to the minima by a significant amount. Among these, DE performs the best, followed by HOPSO and PSO.

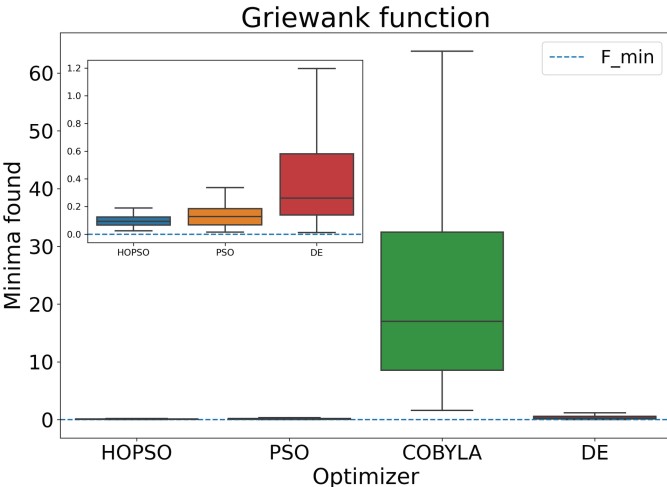

**Fig 11. Performance of the optimizers on Griewank function.** HOPSO and PSO perform better than the other two competitors by being more precise and accurate. As can be seen from the figure, HOPSO is more precise and has a lower median than PSO.

of the $s$ parameter in cases where, in its original settings, it was outperformed by DE. For the Michalewicz function the mean result improved from $-4.5119$ ($s = 10$) to $-4.5860$ ($s = 0.1$) and for the Rastrigin function from $12.458$ ($s = 10$) to $10.841$ ($s = 1$), in both cases outperforming the DE. Detailed results with a varying scaling factor $s$ are shown in Figs 16–17.

## 5 Conclusion

In this paper, we suggest a new optimization method based on the well-researched particle swarm optimization (PSO) method by introducing the concept of conservation of energy to prohibit wild oscillations and velocity explosions. In the newly developed method, we also

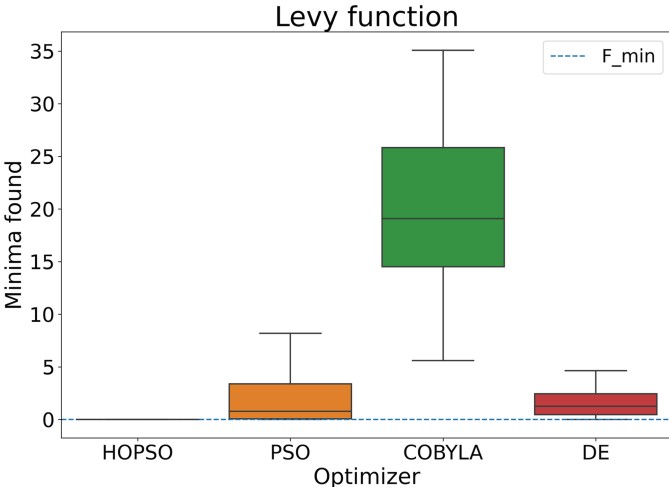

**Fig 12. Performance of the optimizers on Levy function.** Except for HOPSO, all optimizers fail to converge to the minima. HOPSO performs the best followed by PSO, DE and COBYLA.

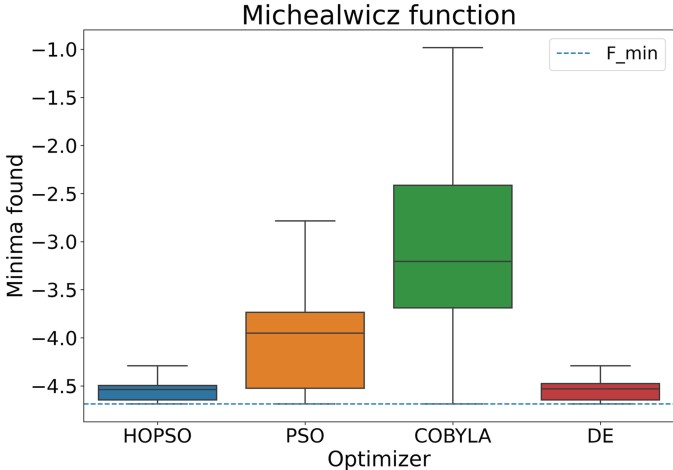

**Fig 13. Performance of the optimizers on Michalewicz function.** HOPSO and DE show the most consistent performance with minimal spread. DE performs the best, slightly ahead of HOPSO, which significantly outperforms both PSO and COBYLA.

introduced an independent tuning parameter that allows one to adjust the convergence of the method depending on the expected number of function evaluations.

We have shown the power of the new procedure through its application onto a set of standard benchmark test optimization functions and compared to the original PSO method, as well as other non-gradient methods including COBYLA and DE. In all cases, HOPSO showed to be at least as good as the original PSO. Thus, in any case where one may consider using PSO for their optimization purposes, HOPSO merits consideration over the original PSO method. HOPSO also outperformed COBYLA in all cases.

In most cases, it has also outperformed the DE method. In some of the cases, it turned out that the tunability of HOPSO allows better results than DE. The few cases where DE proved to

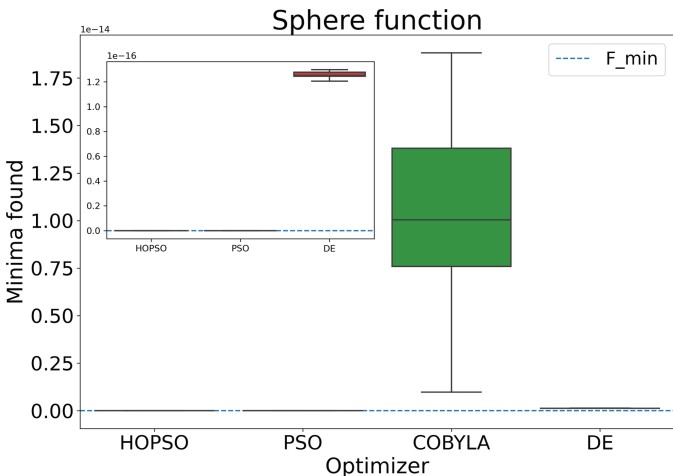

**Fig 14. Performance of the optimizers on sphere function.** All optimizers converge to the minima. For higher accurate scenarios, HOPSO and PSO perform the best.

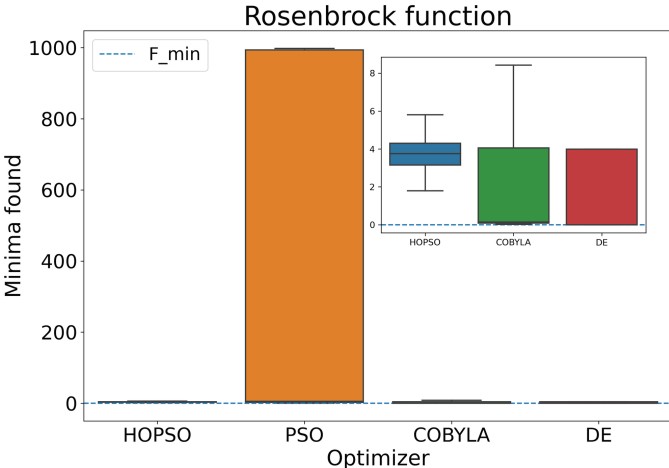

**Fig 15. Performance of the optimizers on Rosenbrock function.** None of the optimizers successfully converge to the minima. However, DE and COBYLA outperform both PSO and HOPSO, with PSO showing the weakest performance by far.

be better are associated with situations where none of the optimizers managed to get a reasonably good result close to the actual minimum – in these cases the DE was able to scan a larger space than any standard method. This is due to the way the DE method works, which allows to search a very wide area of parameters quickly, leading to a fast determination of different possible optima. However, choosing the correct place to "dig deeper" and determine the minimum with reasonable accuracy is another task where other methods, especially HOPSO, showed to be stronger.

HOPSO has a broad range of possible applications. Its robust performance in controlling convergence, ability to avoid local optima, and efficiently explore complex solution spaces make it highly promising for engineering design optimization, where fine-tuning parameters is critical, as well as for neural network training and financial modeling, areas where there

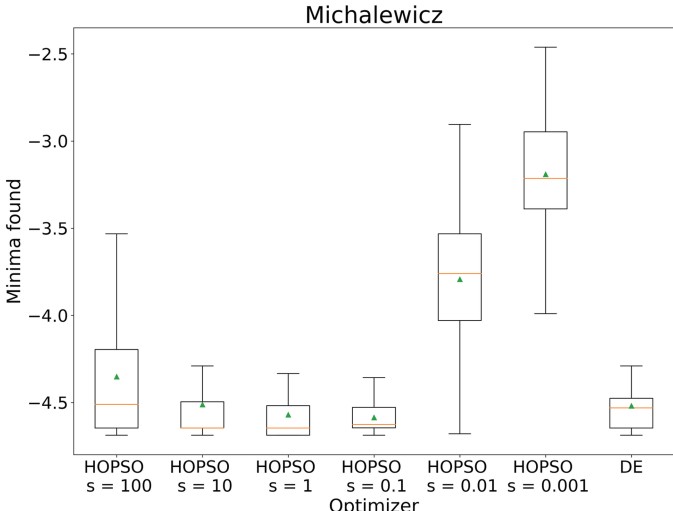

**Fig 16. Performance of the HOPSO optimizer on varying the scaling parameter *s* on Michalewicz function.** The results of HOPSO with scaling factor set to 0.1 and 1 outperform the previously best-performing optimizer, DE.

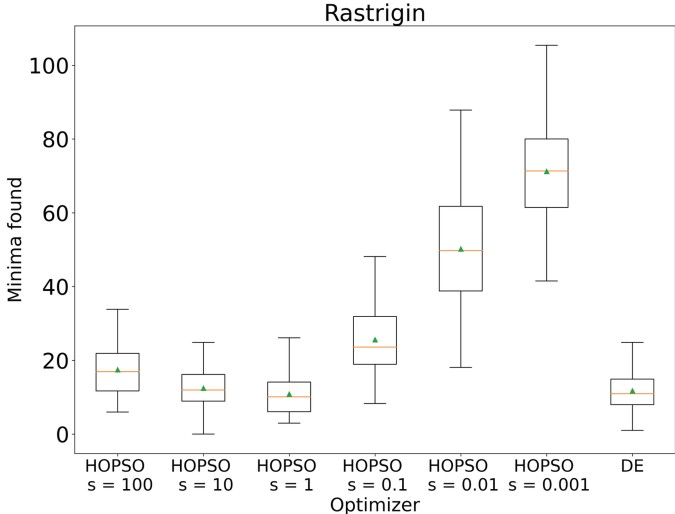

**Fig 17. Performance of the HOPSO optimizer on varying the scaling parameter *s* on Rastrigin function.** The results of HOPSO with the setting of the scaling factor *s* to 1 outperforms the previously best-performing optimizer, DE.

is a high number of tunable parameters and volatile conditions. The balance of exploration and exploitation ensures that the algorithm can be applied effectively to problems in robotics, control systems, and any field where reliable global optimization is essential.

In our future research, we foresee the application of HOPSO in high-dimensional optimization problems that involve a significant number of parameters, in particular in connection with quantum computers and Variational Quantum Algorithms. We believe that the simple tunability of the convergence of the method will allow a better search in these highly degenerate problems.

## Author contributions

**Conceptualization:** Yury Chernyak, Ijaz Ahamed Mohammad, Nikolas Masnicak, Matej Pivoluska, Martin Plesch.

**Data curation:** Yury Chernyak, Ijaz Ahamed Mohammad.

**Formal analysis:** Yury Chernyak, Ijaz Ahamed Mohammad, Nikolas Masnicak, Martin Plesch.

**Investigation:** Martin Plesch.

**Methodology:** Yury Chernyak, Ijaz Ahamed Mohammad, Nikolas Masnicak, Matej Pivoluska, Martin Plesch.

**Project administration:** Martin Plesch.

**Software:** Yury Chernyak, Ijaz Ahamed Mohammad.

**Supervision:** Matej Pivoluska, Martin Plesch.

**Validation:** Yury Chernyak, Ijaz Ahamed Mohammad.

**Visualization:** Yury Chernyak, Ijaz Ahamed Mohammad.

**Writing – original draft:** Yury Chernyak, Ijaz Ahamed Mohammad, Nikolas Masnicak, Matej Pivoluska.

**Writing – review & editing:** Yury Chernyak, Ijaz Ahamed Mohammad, Martin Plesch.

## Supporting information

**S1. COCO benchmarking**
(PDF)

## Acknowledgments

The authors would like to thank Martin Friak and his team for valuable discussions.

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
