## [Decision Letter · Decision Letter 0]

25 Feb 2025

PONE-D-25-05664Harmonic Oscillator based Particle Swarm OptimizationPLOS ONE

Dear Dr. Mohammad,

Thank you for submitting your manuscript to PLOS ONE. After careful consideration, we feel that it has merit but does not fully meet PLOS ONE’s publication criteria as it currently stands. Therefore, we invite you to submit a revised version of the manuscript that addresses the points raised during the review process.

We look forward to receiving your revised manuscript.

Kind regards,

Yuanchao Liu

Academic Editor

PLOS ONE

Additional Editor Comments:

Three reviewers have provided some comments, please revise this manuscript accordingly.

Reviewers' comments:

Reviewer's Responses to Questions

**Comments to the Author**

1. Is the manuscript technically sound, and do the data support the conclusions?

Reviewer #1: Yes

Reviewer #2: Yes

Reviewer #3: Yes

2. Has the statistical analysis been performed appropriately and rigorously? 

Reviewer #1: No

Reviewer #2: N/A

Reviewer #3: Yes

3. Have the authors made all data underlying the findings in their manuscript fully available?

Reviewer #1: Yes

Reviewer #2: Yes

Reviewer #3: No

4. Is the manuscript presented in an intelligible fashion and written in standard English?

Reviewer #1: No

Reviewer #2: Yes

Reviewer #3: Yes

5. Review Comments to the Author

Reviewer #1: In this article, the authors present an extension of the Particle Swarm Optimization (PSO) algorithm. They added a component based on harmonic oscillators to the movement of the particles to prevent both speed explosion and inappropriate stalling. They further evaluated the algorithm on standard benchmark functions and compared its performance to similar algorithms.

Overall, the approach is reasonable and performs well enough. I have, however, a few issues with the paper:

- I found the beginning of the paper well written, but the quality of English quickly drops once we reach Section 1 (which should probably be numbered 2). From this point onwards, many sentences have awkward constructions, some "\ref"s were not correctly compiled (appearing as "??" instead), and we even get an entire paragraph duplicated (pre- and post-editing, I assume). The whole text requires that kind of attention before publication.

- The structure of the paper itself is strange. The introduction section has no number and many elements that would be considered methods or discussion (e.g., lines 433-450 where the authors discussed a parameter of their algorithm).

- I did appreciate the didactic approach of the authors, providing a lot of background about the field and PSO in particular. However, their review of the field (page 2) feels a bit dated. In particular, there was no mention of Quality-Diversity algorithms, which have become fairly popular over the past 10 years. Moreover, there are multiple ways to categorize algorithms. While I have seen similar splits in the past, I feel like the authors would be better served citing a review of the field rather than providing ~30 references to mostly unrelated algorithms.

- The results focus only on the average performance of the algorithms at the end of a run, with no evaluation of convergence speed. Yet, the whole point of the proposed algorithm was to help convergence. Considering that the authors are using standard benchmark functions, I suggest that they use the COCO (COmparing Continuous Optimizers) platform to evaluate how well their algorithm performs. That platform does not only look at how well algorithms performed, it provides insights on how fast and how reliably they got to the solution. Plus, evaluation data for *many* algorithms are already available, saving the authors the effort of running the platform on other algorithms (see https://coco-platform.org/testsuites/bbob/data-archive.html which includes PSO, variants of it, as well as the implementation of COBYLA and DE used by the authors).

Overall, I think this article is promising but still requires some effort before being publishable.

Minor comments:

- The abstract spends too many words on basic information about the field and PSO. The authors should cut some of it and instead provide more details about their approach and results.

- "Introduction" should have a number.

- Lines 153-154: "their sum ... directly impacts" -> *should* directly impact. The authors are still establishing what \chi should be. Alternatively, they could move equation (3) after line 151 instead, explaining that it is an established version of the parameter, *then* describe its formula.

- Line 164: The authors mention that c=2.05 is a common setting, but do not provide any further comment on that value. While they provide a reference, I would appreciate it if they could provide some insights on its impact.

- Line 208-209: "one optimally needs to have both singular values similar, and for smooth convergence it must be [...]" -> what is "it"? I assume that the authors mean the average, based on further elements.

- Line 225: "making the explosions less probable." The results suggest that a high r makes the explosions *more* probable. I assume that was a typo.

- Line 226: "The results [...] have been confirmed" -> provide a citation.

- Lines 235: "This understanding is also confirmed" -> provide a citation.

- Lines 427-428: The authors should mention that the budget is available in Table 3.

- Line 457: "not apparent in the figure" -> then what forms the basis for the authors' conclusion? (Note that this problem will probably be solved by using COCO)

- Lines 469-470: "more precise and more accurate" What do the authors mean by "precise" in this case?

- Lines 531-532: the authors mention again that they introduced a concept of energy conservation, but that is not really the case (especially since energy may be lost over time). I think they should rephrase this sentence to reflect better what they did.

Reviewer #2: This paper proposes a new hybrid optimization algorithm, particle swarm optimization (HOPSO) of harmonic oscillator. By combining particle swarm optimization (PSO) with the physical principle of harmonic oscillator, the convergence control problem of traditional PSO is solved. The author constructs a dynamic model based on damping harmonic oscillator from the theoretical level, and introduces energy conservation and controllable damping mechanism. Redefined the equation of motion of particles.By setting the amplitude threshold, HOPSO achieves fine control of particle velocity and search range. The experimental part compares the performance of HOPSO with PSO, COBYLA and DE in 12 standard test functions (covering single mode, multi-mode, high-dimensional, low-dimensional and other scenarios), and verifies its superiority in most scenarios, especially in complex multi-modal functions and high-dimensional problems. The theoretical explanation is relatively complete, but further optimization is needed in the aspects of overview, comparison of results and speech norms:

1. The introduction part describes redundancy (a large amount of repeated description of PSO problems), which needs to be streamlined and optimized. It is suggested to highlight the significance of HOPSO research, especially the uniqueness and advantages of HOPSO algorithm in modern optimization problems.

2. It is necessary to strengthen the comprehensiveness of the literature review, especially the insufficient discussion on the existing physically-inspired optimization algorithms (such as GSA, BBBC, etc.), which fails to fully position HOPSO's innovation in the field. It is suggested to simplify the summary of existing PSO methods, and clarify the innovation and contribution of this study.

3. The explanation of the energy conservation mechanism is insufficient. The illustration in line 138 is unclear; The part of the explanation of harmonic motion about the virtual mass of the particle and the stiffness of the spring is more intuitive with a physical diagram of the relationship.

4. The presentation of the results is missing, and the overall quality of the image needs to be optimized, which is not rigorous enough. FIG. 4 and FIG. 6 The abscissa of the small figure is missing.

5. The lack of in-depth analysis of the reasons why DE performs better in some scenarios weakens the comprehensiveness of the conclusion.

6. The format of the reference part is not uniform, such as the reference [45], which needs to be adjusted according to the requirements of the journal.

7. Further emphasize the practical application prospect of the algorithm.

8. The authors need to inspire class related algorithms and applications for more discussion, for example, DOI: 10.1109 / TCBB. 2020.3011582, DOI: 10.1109 / TNB. 2021.3056351, DOI: 10.1109 / JBHI. 2025.3541848.

Reviewer #3: Harmonic Oscillator based Particle Swarm Optimization

Reviewer Comments

This paper is well-written and nicely introduces the PSO and subsequent HOPSO algorithms. If this paper is published, I would seriously consider posting the code used to generate the figures, as it would be good to allow people to verify the results in this work and possibly use the algorithms themselves.

Major Points

1. Line 333, what is the range of the time? I’m assuming this is the time of the harmonic oscillator. This could be a minor point, but the dynamics change considerably depending on the range chosen.

2. The minima of the functions should be defined. That is, what do the minima values represent in terms of error within the function?

Minor Points

1. Line 136, figure ??

2. Line 138, Table ??

3. Label on figure 2 y-axis.

4. Line 216, “Singular values…” sentence should be reworded.

5. Line 217, be more specific with what the region is.

6. The paragraph 247-2 is a repeat of the previous paragraph, just reworded.

7. Line 304, more specifically an underdamped harmonic oscillator

8. Equation 14 x(t) is not equal to A_0 (-e^(λ (-t))) (λ cos(θ + t ω) since you lose the x_0 term.

9. Line 332/333 end of sentence …A_0 decays as… needs to be rewritten expressly stating A_0 (e^(λ (-t))) decays as A_0 (e^(λ (-t))) moves forward in time? If A 0 is decaying not the overall term A_0 (e^(λ (-t))) state how.

10. Paragraph 335-340 should be used after equation (20) if equation (20) is going to be referenced in this way.

11. Table 3 appears in the middle of the References

12. In algorithm 1, the line if personal best value < global best energy then, was if personal best value < global best value meant to be the line?

13. The figures comparing the optimization functions would benefit from titles with the function the algorithms are trying to optimize.

14. Figure 6, DE label is missing from small inner figure.

15. In some of the figures (4,5,6), it is stated that HOPSO has the highest precision, you would need to zoom in enough to show this in the figure or refer readers to Table 3 instead.

16. In the figures 4-15, a line drawn at the true F_min would be helpful.

1. State the programming language this was written in and figures produced from.

6. PLOS authors have the option to publish the peer review history of their article (what does this mean?). If published, this will include your full peer review and any attached files.

Reviewer #1: No

Reviewer #2: No

Reviewer #3: **Yes: **Trevor Reckell

---

## [Author Response · Author response to Decision Letter 1]

14 Apr 2025

Dear Editorial team,

We sincerely appreciate the reviewers' thoughtful and constructive comments on our manuscript, "Harmonic Oscillator based Particle Swarm Optimization" (PONE-D-25-05664). We have carefully considered each suggestion and revised the manuscript accordingly. Below, we provide a detailed response to each comment. Changes made in the manuscript are written in red for clarity.

Reviewer 1:

Comment 1: I found the beginning of the paper well written, but the quality of English quickly drops once we reach Section 1 (which should probably be numbered 2) . From this point onwards, many sentences have awkward constructions, some "\ref"s were not correctly compiled (appearing as "??" instead) , and we even get an entire paragraph duplicated (pre- and post-editing, I assume) . The whole text requires that kind of attention before publication.

Response : Thank you for the observation. All the sections are now numbered and the references are compiled properly. We also took care about some complicated formulations and occasional repetitiveness.

Comment 2: The structure of the paper itself is strange. The introduction section has no number and many elements that would be considered methods for discussion (e.g., lines 433-450 where the authors discussed a parameter of their algorithm).

Response: We have numbered the introduction section. We also did add several subsections to structure the text in a better way. We believe that in the present form it is easier to read and more accessible.

Comment 3: I did appreciate the didactic approach of the authors, providing a lot of background about the field and PSO in particular. However, their review of the field (page 2) feels a bit dated. In particular, there was no mention of Quality-Diversity algorithms, which have become fairly popular over the past 10 years. Moreover, there are multiple ways to categorize algorithms. While I have seen similar splits in the past, I feel like the authors would be better served citing a review of the field rather than providing ~30 references to mostly unrelated algorithms.

Response: We have included additional information on Quality-Diversity algorithms along with relevant references. Regarding the literature review references, we prefer to include original references to different works. However, as suggested, we have also added a few relevant review articles that discuss various optimization algorithms.

Comment 4: The results focus only on the average performance of the algorithms at the end of a run, with no evaluation of convergence speed. Yet, the whole point of the proposed algorithm was to help convergence. Considering that the authors are using standard benchmark functions, I suggest that they use the COCO (COmparing Continuous Optimizers) platform to evaluate how well their algorithm performs. That platform does not only look at how well algorithms performed, it provides insights on how fast and how reliably they got to the solution. Plus, evaluation data for *many* algorithms are already available, saving the authors the effort of running the platform on other algorithms (see https://coco-platform.org/testsuites/bbob/data-archive.html which includes PSO, variants of it, as well as the implementation of COBYLA and DE used by the authors).

Response: The reviewer brings up a very interesting point. Let us stress that while speaking about convergence, we always meant the ability of the method to find the minimum with good precision within a given budget specified in advance. As parameters of the optimization method (such as damping) do depend on this budget, it does not make much sense to look at how quick, while running, the method identifies the optimum, as if we would aim for a shorter run (i.e. lower budget), we would have set a different value of parameters.

However, as suggested, we did analysis using COCO and compared it with our implementation of PSO, scipy’s implementation of COBYLA and DE. COCO post processing analysis was performed for all test functions using 15 instances with the maximum evaluations budget of 10000*dimension. A summary of results is shown in the figures below for 2,3,5,10 and 20 dimensions respectively in the cover letter.

The results show that HOPSO consistently outperforms the other algorithms, achieving higher success rates with fewer function evaluations per dimension. However, we have decided to omit this COCO analysis from the article, as our primary goal is to aim on the success rate while using the whole budget.

Minor comments:

5. The abstract spends too many words on basic information about the field and PSO. The authors should cut some of it and instead provide more details about their approach and results.

Response: The abstract was shortened.

6. "Introduction" should have a number.

Response: Done

7. Lines 153-154: "their sum ... directly impacts" -> *should* directly impact. The authors are still establishing what \chi should be. Alternatively, they could move equation (3) after line 151 instead, explaining that it is an established version of the parameter, *then* describe its formula.

Response: Done

8. Line 164: The authors mention that c=2.05 is a common setting, but do not provide any further comment on that value. While they provide a reference, I would appreciate it if they could provide some insights on its impact.

Response: Reference was added. The larger the value is, the stronger the attraction towards the attractor is. Value slightly larger than 2, multiplied by a random number that is on average 0,5, results in a value just over one leading to (almost) equal weight of the inertia of the particle (current velocity) and the attraction force. The constriction factor defines then the damping result in convergence.

9. Line 208-209: "one optimally needs to have both singular values similar, and for smooth convergence it must be [...]" -> what is "it"? I assume that the authors mean the average, based on further elements.

Response: As this part of the manuscript was really hard to read, a few paragraphs were rewritten.

10. Line 225: "making the explosions less probable." The results suggest that a high r makes the explosions *more* probable. I assume that was a typo.

Response: As this part of the manuscript was really hard to read, a few paragraphs were rewritten.

11. Line 226: "The results [...] have been confirmed" -> provide a citation.

Response: Done

12. Lines 235: "This understanding is also confirmed" -> provide a citation.

Response: Done

13. Lines 427-428: The authors should mention that the budget is available in Table 3.

Response: Done

14. Line 457: "not apparent in the figure" -> then what forms the basis for the authors' conclusion? (Note that this problem will probably be solved by using COCO)

Response: We have explained the fact that while not every feature is apparent in the plot due to limited zoom, everything can be accessed in the table with numerical results.

15. Lines 469-470: "more precise and more accurate" What do the authors mean by "precise" in this case?

Response: We consistently use the word accurate in the manuscript now.

16. Lines 531-532: the authors mention again that they introduced a concept of energy conservation, but that is not really the case (especially since energy may be lost over time). I think they should rephrase this sentence to better reflect what they did.

Response: We have added a paragraph introducing the concept of energy conservation in conservative force fields supplemented by damping via friction forces.

Reviewer 2:

Comment 1: The introduction part describes redundancy (a large amount of repeated description of PSO problems), which needs to be streamlined and optimized. It is suggested to highlight the significance of HOPSO research, especially the uniqueness and advantages of HOPSO algorithm in modern optimization problems.

Response: We have incorporated a paragraph in the introduction the specific advantages of HOPSO and how it compares to PSO thereby indicating the significance for the HOPSO research.

Comment 2: It is necessary to strengthen the comprehensiveness of the literature review, especially the insufficient discussion on the existing physically-inspired optimization algorithms (such as GSA, BBBC, etc.), which fails to fully position HOPSO's innovation in the field. It is suggested to simplify the summary of existing PSO methods, and clarify the innovation and contribution of this study.

Response: We have elaborated the physically inspired algorithms paragraph a little more and added relevant references, as well as explained the applications of HOPSO at the end of the manuscript.

Comment 3: The explanation of the energy conservation mechanism is insufficient. The illustration in line 138 is unclear; The part of the explanation of harmonic motion about the virtual mass of the particle and the stiffness of the spring is more intuitive with a physical diagram of the relationship.

Response: A more thorough explanation of harmonic motion and its workings to model this optimization method was added in section 3.

Comment 4: The presentation of the results is missing, and the overall quality of the image needs to be optimized, which is not rigorous enough. FIG. 4 and FIG. 6 The abscissa of the small figure is missing.

Response: All figures have been revised to include the F_min line and enhanced boxplot visibility through extensive zooming where necessary.

Comment 5: The lack of in-depth analysis of the reasons why DE performs better in some scenarios weakens the comprehensiveness of the conclusion.

Response: Our aim in the manuscript was to further develop the PSO method by including the physical concept of energy and its controlled dissipation. DE was introduced solely to compare the outputs of the HOPSO method and we did not analyze in detail the exact working of the DE method. As in DE, the new positions are constructed by mixing different old positions per dimensions, it is very hard to track why a specific new position appears. As such, the randomness aspect in this method is much stronger than in physics-based methods, which on one hand can lead to good performance in unknown and intractable scenarios, but on the other hand it makes it very hard, if not impossible to actually understand the decision process of the method.

Comment 6: The format of the reference part is not uniform, such as the reference [45], which needs to be adjusted according to the requirements of the journal.

Response: We have corrected the format for the reference.

Comment 7: Further emphasize the practical application prospect of the algorithm.

Response: Practical applications and utility of HOPSO has now been addressed in the conclusion paragraph.

Comment 8: The authors need to inspire class related algorithms and applications for more discussion, for example, DOI: 10.1109 / TCBB. 2020.3011582, DOI: 10.1109 / TNB. 2021.3056351, DOI: 10.1109 / JBHI. 2025.3541848.

Response: We have incorporated relevant references to illustrate the practical applications of these optimization methods across various domains.

Reviewer 3:

Comment 1: Line 333, what is the range of the time? I’m assuming this is the time of the harmonic oscillator. This could be a minor point, but the dynamics change considerably depending on the range chosen.

Response: We have explained this in detail in the manuscript. For small ranges, the range itself matters. For larger ranges, as this is a harmonic motion, the range stops to be relevant, especially if taken as a multiple of Pi.

Comment 2: The minima of the functions should be defined. That is, what do the minima values represent in terms of error within the function?

Response: We have clearly displayed the optimal minima of the functions in the result table.

Minor comments:

1. Line 136, figure ??

2. Line 138, Table ??

3. Label on figure 2 y-axis.

4. Line 216, “Singular values…” sentence should be reworded.

5. Line 217, be more specific with what the region is.

6. The paragraph 247-2 is a repeat of the previous paragraph, just reworded.

7. Line 304, more specifically an underdamped harmonic oscillator

8. Equation 14 x(t) is not equal to A_0 (-e^(λ (-t))) (λ cos(θ + t ω) since you lose the x_0 term.

9. Line 332/333 end of sentence …A_0 decays as… needs to be rewritten expressly stating A_0 (e^(λ (-t))) decays as A_0 (e^(λ (-t))) moves forward in time? If A 0 is decaying not the overall term A_0 (e^(λ (-t))) state how.

10. Paragraph 335-340 should be used after equation (20) if equation (20) is going to be referenced in this way.

11. Table 3 appears in the middle of the References

12. In algorithm 1, the line if personal best value < global best energy then, was if personal best value < global best value meant to be the line?

13. The figures comparing the optimization functions would benefit from titles with the function the algorithms are trying to optimize.

14. Figure 6, DE label is missing from small inner figure.

15. In some of the figures (4,5,6), it is stated that HOPSO has the highest precision, you would need to zoom in enough to show this in the figure or refer readers to Table 3 instead.

16. In the figures 4-15, a line drawn at the true F_min would be helpful.

17. State the programming language this was written in and figures produced from.

Response towards minor comments:

We have incorporated all the changes as suggested.

We appreciate the reviewers’ insightful feedback, which has helped us improve our manuscript. We believe the revisions address the concerns raised and enhance the clarity of our work. We hope the revised manuscript is now suitable for publication.

Best regards,

The authors

---

## [Decision Letter · Decision Letter 1]

8 May 2025

PONE-D-25-05664R1Harmonic Oscillator based Particle Swarm OptimizationPLOS ONE

Dear Dr. %LAST_NMAE%,

Thank you for submitting your manuscript to PLOS ONE. After careful consideration, we feel that it has merit but does not fully meet PLOS ONE’s publication criteria as it currently stands. Therefore, we invite you to submit a revised version of the manuscript that addresses the points raised during the review process.

We look forward to receiving your revised manuscript.

Kind regards,

Yuanchao Liu

Academic Editor

PLOS ONE

Additional Editor Comments :

One reviewer still has some comments, please address.

Reviewers' comments:

Reviewer's Responses to Questions

**Comments to the Author**

1. If the authors have adequately addressed your comments raised in a previous round of review and you feel that this manuscript is now acceptable for publication, you may indicate that here to bypass the “Comments to the Author” section, enter your conflict of interest statement in the “Confidential to Editor” section, and submit your "Accept" recommendation.

Reviewer #1: (No Response)

Reviewer #2: All comments have been addressed

2. Is the manuscript technically sound, and do the data support the conclusions?

Reviewer #1: (No Response)

Reviewer #2: Yes

3. Has the statistical analysis been performed appropriately and rigorously? 

Reviewer #1: (No Response)

Reviewer #2: N/A

4. Have the authors made all data underlying the findings in their manuscript fully available?

Reviewer #1: (No Response)

Reviewer #2: Yes

5. Is the manuscript presented in an intelligible fashion and written in standard English?

Reviewer #1: (No Response)

Reviewer #2: Yes

6. Review Comments to the Author

Reviewer #1: I have read the response from the authors and the edited paper. The authors have addressed most of my concerns, but I still have some comments.

The authors added a short part on QD algorithms, but I disagree with its content.

"A rather specific branch of optimization methods are Quality diversity (QD)"

> But it still aims to optimize a single cost function, so it should belong to the first category (specifically to the "population-based/evolutionary" subcategory). While the authors of reference [2] claim that QD is a different branch of stochastic optimization, that is not the case based on the classification presented in the current paper. I am fine with them calling it a specific case, though (as long as it is in the proper category, or at least comes later than the general case).

"rather than optimizing toward a single global optimum." 

> QD algorithms are still looking for the global optimum (if it exists). Exploring the feature space is, among other things, a way to avoid local traps. See Figure 1 in reference [2].

"They work in behavioral space (feaure space) rather than the genotypic (parameter) space. "

> That sentence (note: "feature", not "feaure") comes from the abstract of [2], but it seems that the author misunderstood its meaning. To be fair, I find that sentence extremely misleading in the original article. Here, Chatzilygeroudis et al. mean that the selection operator will work on the collection of solutions currently stored, across different features, rather than just select based on the best fitness in the current solutions. It is clear from looking at their section "4.2 Connections to Multimodal Optimization" that QD algorithms do evolve solutions in the genotypic (parameter) space.

"all nicely summarized in [4], demonstrating their versatility in tackling

complex, multimodal optimization tasks [5, 6]."

> As far as I can tell, [4] does not mention QD algorithms. [2] and [5] do provide an overview of QD algorithms, though. The authors may have mixed up their references.

My main concern is with comment 4.

I could not find the results the authors mention anywhere, so I cannot comment on them. However, it seems strange to me to exclude those results, especially if they agree with the authors’ premise. Additionally, the dynamics of the algorithm are the main selling point of their algorithm, so I would expect to see what happens to the population over time (especially compared to the other algorithms). Figure 3 provides a toy example for a single particle, but it’s not enough. COCO is one way to get information about the population over time, but I would be open to other suggestions. In any case, I would recommend adding those results as supplementary materials if the authors feel they would clutter the main text.

I understand that parameters will change with budget, but that can be simply tackled by running a few versions of the algorithm (using different parameters) for the highest budget. Hopefully, the version of the algorithms that was tuned for a given parameter range will outperform the others over that range. The overall results will provide more insights into the connection between parameters and budget.

As a final minor comment, I checked the code provided by the authors, but it had no readme and no examples. I would recommend providing scripts allowing users to test the results mentioned in the main text.

Reviewer #2: Thanks for Author's response. All my concerns is addressed. So i recommend this paper published in Plos one.

7. PLOS authors have the option to publish the peer review history of their article (what does this mean?). If published, this will include your full peer review and any attached files.

Reviewer #1: No

Reviewer #2: No

---

## [Author Response · Author response to Decision Letter 2]

22 May 2025

Dear Reviewers,

We appreciate your thoughtful and constructive comments on our manuscript, "Harmonic Oscillator based Particle Swarm Optimization" (PONE-D-25-05664). We have carefully considered suggestions and revised the manuscript accordingly. Below, we provide a detailed response to each suggestion. Changes made in the manuscript are written in red for clarity.

Reviewer #1

1) We have revised the paragraph to clarify that QD algorithms do aim to optimize a single objective function, but they also focus on finding diverse solutions across a feature space. We corrected the earlier confusion about where QD algorithms operate–they evolve solutions in parameter space while organizing them in feature space. We also fixed the typo (“feaure” → “feature”) and updated the references to cite only those that actually discuss QD methods. We appreciate the comments, which helped us improve the accuracy and clarity of this section.

2) As for the COCO analysis, we included all the results in the previous rebuttal letter.

As suggested, we have decided to include the COCO-graphs in the supplementary material without changing the structure of the main article.

3) As for the code published on Github, we stress that the code has only a form of a supplementary material for this manuscript - this is now clearly mentioned in the readme file.

Faithfully

The Authors

---

## [Decision Letter · Decision Letter 2]

27 May 2025

Harmonic Oscillator based Particle Swarm Optimization

PONE-D-25-05664R2

Dear Dr. Ijaz Ahamed Mohammad,

We’re pleased to inform you that your manuscript has been judged scientifically suitable for publication and will be formally accepted for publication once it meets all outstanding technical requirements.

Kind regards,

Yuanchao Liu

Academic Editor

PLOS ONE

Additional Editor Comments (optional):

Reviewers' comments:

Reviewer's Responses to Questions

**Comments to the Author**

1. If the authors have adequately addressed your comments raised in a previous round of review and you feel that this manuscript is now acceptable for publication, you may indicate that here to bypass the “Comments to the Author” section, enter your conflict of interest statement in the “Confidential to Editor” section, and submit your "Accept" recommendation.

Reviewer #1: All comments have been addressed

Reviewer #2: All comments have been addressed

2. Is the manuscript technically sound, and do the data support the conclusions?

Reviewer #1: (No Response)

Reviewer #2: Yes

3. Has the statistical analysis been performed appropriately and rigorously? 

Reviewer #1: (No Response)

Reviewer #2: N/A

4. Have the authors made all data underlying the findings in their manuscript fully available?

Reviewer #1: (No Response)

Reviewer #2: Yes

5. Is the manuscript presented in an intelligible fashion and written in standard English?

Reviewer #1: (No Response)

Reviewer #2: Yes

6. Review Comments to the Author

Reviewer #1: (No Response)

Reviewer #2: The author has solved all the problems I had in the last round. I recommend this paper publish in Plos one.

7. PLOS authors have the option to publish the peer review history of their article (what does this mean?). If published, this will include your full peer review and any attached files.

Reviewer #1: No

Reviewer #2: No

---

## [Editor Report · Acceptance letter]

PONE-D-25-05664R2

PLOS ONE

Dear Dr. Mohammad,

I'm pleased to inform you that your manuscript has been deemed suitable for publication in PLOS ONE. Congratulations! Your manuscript is now being handed over to our production team.

Kind regards,

on behalf of

Dr. Yuanchao Liu

Academic Editor

PLOS ONE